# Melatonin in Dermatologic Allergic Diseases and Other Skin Conditions: Current Trends and Reports

**DOI:** 10.3390/ijms24044039

**Published:** 2023-02-17

**Authors:** Iva Bešlić, Liborija Lugović-Mihić, Alen Vrtarić, Ante Bešlić, Ivana Škrinjar, Milena Hanžek, Danijel Crnković, Marinko Artuković

**Affiliations:** 1Department of Dermatovenereology, University Hospital Center Sestre Milosrdnice, 10000 Zagreb, Croatia; 2School of Dental Medicine, University of Zagreb, 10000 Zagreb, Croatia; 3Department of Clinical Chemistry, University Clinical Hospital Center Sestre Milosrdnice, 10000 Zagreb, Croatia; 4Faculty of Dental Medicine and Health, Josip Juraj Strossmayer University, 31000 Osijek, Croatia; 5Department of Oral Medicine, University Hospital Centre, 10000 Zagreb, Croatia; 6Department of Psychiatry, University Hospital Center Sestre Milosrdnice, 10000 Zagreb, Croatia; 7Department of Polemology, Special Hospital for Pulmonary Diseases, 10000 Zagreb, Croatia

**Keywords:** melatonin, atopic dermatitis, urticaria, melasma, rosacea, androgenic alopecia, photoprotection, skin aging, sleep disorders, allergy, allergic rhinitis

## Abstract

Melatonin is the main hormone that regulates the sleep cycle, and it is mostly produced by the pineal gland from the amino acid tryptophan. It has cytoprotective, immunomodulatory, and anti-apoptotic effects. Melatonin is also one of the most powerful natural antioxidants, directly acting on free radicals and the intracellular antioxidant enzyme system. Furthermore, it participates in antitumor activity, hypopigmentation processes in hyperpigmentary disorders, anti-inflammatory, and immunomodulating activity in inflammatory dermatoses, maintaining the integrity of the epidermal barrier and thermoregulation of the body. Due predominantly to its positive influence on sleep, melatonin can be used in the treatment of sleep disturbances for those with chronic allergic diseases accompanied by intensive itching (such as atopic dermatitis and chronic spontaneous urticaria). According to the literature data, there are also many proven uses for melatonin in photoprotection and skin aging (due to melatonin’s antioxidant effects and role in preventing damage due to DNA repair mechanisms), hyperpigmentary disorders (e.g., melasma) and scalp diseases (such as androgenic alopecia and telogen effluvium).

## 1. Introduction

Melatonin is a hormone that plays many useful roles in organisms and can also be used for therapeutic purposes.

It is one of the most powerful natural antioxidants, directly acting on free radicals and the intracellular antioxidant enzyme system [1]. It also has cytoprotective, immunomodulatory, and anti-apoptotic effects [2]. One of melatonin’s most important roles in humans is the initiation and maintenance of the sleep cycle. Due predominantly to this positive influence on sleep, melatonin can be used as a therapeutic supplement and in the treatment of various diseases accompanied by sleep disturbances.

It has been observed that dermatology is a field in which melatonin is currently rarely used; however, recently, melatonin has been gradually gaining importance, which is supported by numerous studies yielding positive results for its clinical use in diseases such as atopic dermatitis (AD), hyperpigmentation disorders, and scalp disorders [3]. In addition, the latest research has also investigated the use of melatonin in the prevention of skin tumors, photoprotection, and stimulating skin regeneration, which is particularly important in the prevention of aging [3].

## 2. The Melatonin Molecule and Its Effects on the Human Body

Melatonin (N-acetyl-5-methoxytryptamine) is a hormone that plays a central role in the sleep cycle. It is mostly produced by the pineal gland from the amino acid tryptophan, and it is secreted into the blood and cerebrospinal fluid [1,4]. Other tissues aside from the epiphysis also produce melatonin, including the retina, bone marrow, gonads, and gastrointestinal mucous, as well as the skin, but the role of melatonin in these tissues is still largely unknown [1].

The process of melatonin synthesis is complex. The pineal gland is a neuroendocrine gland located in the brain that contains melatonin-producing cells (pinealocytes). The synthesis of melatonin begins with the amino acid tryptophan, which, with the action of the enzyme tryptophan hydroxylase, is transformed into 5-hydroxytryptophan. This is then transformed into serotonin. Serotonin is acetylated (with the enzyme arylalkylamine N-acetyltransferase) into N-acetylserotonin, which is then converted into melatonin [1]. The process of melatonin synthesis in the pinealocytes of the pineal gland is under the control of the suprachiasmatic paraventricular nuclei of the hypothalamus; however, melatonin synthesis is primarily controlled by the circadian system rhythm, by which melatonin is produced daily in synchronization with the light-dark cycle [1].

In 1975, it was discovered that the production of melatonin in humans follows the circadian rhythm and that nighttime concentrations of plasma melatonin are ten times higher than daytime concentrations [1]. Light (most often from the blue spectrum) inhibits melatonin synthesis in such a way that it activates the breakdown of melanopsin (a photopigment that absorbs light in retinal ganglion cells) and inhibits melatonin synthesis via the retinohypothalamic pathway [5,6]. Darkness leads to the activation of postganglionic sympathetic neurons, which affects the secretion of noradrenaline and enables the further synthesis of melatonin. At the same time, the enzyme that participates in the synthesis of melatonin, N-acetyl-transferase, is activated [1]. Therefore, melatonin is not stored inside the pinealocytes and is released in the form in which it was synthesized. Melatonin is a lipophilic molecule that easily diffuses into the cerebrospinal fluid of the central nervous system during the night, as well as into the bloodstream. In the bloodstream, melatonin is mostly bound to albumins (approximately 70% of total melatonin), and a smaller part of the molecule is found in its free form [7]. In the liver, melatonin is metabolized into 6-hydroxymelatonin by cytochrome P450 and conjugated into 6-sulfatoxymelatonin, which is subsequently excreted through urine [8].

In humans, melatonin achieves its effects via G-protein coupled membrane receptors, nuclear receptors, calmodulin, and antioxidant properties. Membrane receptors are located in the cerebrum and peripheral organs (spleen, thymus, lymphocyte cells, etc.) [2]. There are two membrane receptor types: the high-affinity Mel1a receptors (ML1, ML1a, MT1, and MTNR1A) (which are primarily located at various sites in the brain and skin) and low-affinity affinity Mel1b receptors (MT2, ML1b, and MTNR1B), also located at various sites in the brain [2]. Concerning signaling, post-receptor signaling mechanisms are performed via the inhibition of adenylate cyclase and the reduction of cAMP, as well as Mel1b receptors inhibiting guanylate cyclase and reducing cGMP [2]. Regarding nuclear receptors, there are two receptor types: the orphan receptor retinoid Z receptor (RZR)- β and the retinoic acid-related orphan receptor (ROR)-α, β, and γ [2]. They are necessary for the mast cells’ function and their role in inflammation, immune response, cell proliferation, and apoptosis mechanisms [2]. Moreover, they can be associated with the transcription of nuclear factor-κB (NF-κB) as the post-receptor signaling pathway [2]. Genetic processes are very important for these mechanisms. During inflammation, NF-κB induces endogenous melatonin synthesis from inflammatory cells for mast cell regulation (by stimulating the arylalaminamine-acetyltransferase enzyme) [2]. Stimulation of mast cells is important for melatonin to have an effect: While melatonin administration in the presence of unstimulated mast cells does not affect the endogenous melatonin level, in the presence of stimulated mast cells, it inhibits NF-κB and reduces the endogenous melatonin level in a dose-dependent manner. It could be a mechanism that can stop inflammatory responses [2].

As mentioned above, melatonin has significant antioxidant and cytoprotective effects. It is a free radical scavenger and an antioxidant that stimulates superoxide dismutase, glutathione peroxidase, and glutathione reductase enzymes. It also neutralizes molecules such as hydrogen peroxide, oxygen radicals, peroxynitrite anion, nitric oxide, and hypochloride acid [2].

Melatonin also has significant immunomodulatory effects. Its receptors are also present in human lymphocytes, and as explained in the literature data, lymphocytes synthesize melatonin, secrete it, and respond to melatonin [2]. Melatonin also participates in T cell differentiation and activation and promotes the production of IFN-γ and IL-2 (via the membrane and nuclear receptors); thus, it may activate human Th1 cells [2]. As an immunomodulatory hormone, it influences Th1, Th2, Th17, and Treg responses, in a different manner—in the case of immunosuppression, melatonin inhibits the Th1, Th17, and Treg responses, while in the case of immune exacerbation, it stimulates the Treg pathways [2]. Its immunomodulatory mechanism allows the production of various cytokines (such as IL-1, IL-2, IL-6, and IL-10) and it increases T cell activity and regulates cell proliferation; thus, it indirectly increases antibody production. Increased IFN-γ and IL-2 create a positive feedback loop for the synthesis of melatonin and IL-12 (increased IL-2 levels lead to increased Natural Killer (NK) activity) [2].

Melatonin serves an antiapoptotic effect purpose, as has been confirmed in breast cancer. Calmodulin-mediated processing is often associated with breast cancer development, and it has been suggested that melatonin facilitates dephosphorylation and nuclear import of histone deacetylase 4, leading to the inactivation of calmodulin-dependent protein kinase II alpha and apoptosis [2].

Due to the many different roles that it plays in the body, there are also (three) different ways in which melatonin values can be measured/assessed for research purposes—in the blood, urine, or saliva [9,10]. In the blood, the highest melatonin concentrations are reached between 00:00 h and 05:00 h at night, after which they begin to fall again [9]. In the urine, values of 6-sulfatoxymelatonin reflect melatonin levels in plasma, which allows the melatonin concentration in urine to be measured by a less invasive method that is also reliable for assessing both pineal function and melatonin production [8]. Melatonin can also be excreted in saliva, but only in the free form of melatonin that is not bound to albumins [8]. When measuring melatonin, it is important to consider exactly when and how it is measured. According to research the deviation between salivary melatonin and plasmatic melatonin can vary as much as 36% [10]. The concentration of salivary melatonin also depends on the part of the day in which it is measured—values range between 1 and 5 pg/mL, while nighttime values can be between 10 and 50 pg/mL [9].

## 3. Melatonin as a Therapeutic Option

Melatonin is an important physiological sleep regulator, and adequate production is very important for achieving good-quality sleep. The most common sleep-related disorders include difficulty falling asleep, early waking, and a feeling of fatigue that disrupts daily activities and consequently leads to difficulties in the individual’s work and social life [11,12]. Several studies have shown that the exogenous intake of melatonin for diseases accompanied by sleep disorders increases the body’s concentrations of melatonin and favorably affects the quality of sleep; therefore, the systemic application of melatonin for sleep disorders has become generally accepted [13,14,15,16,17,18,19]. Factors that have a negative effect on melatonin production are aging, the presence of certain diseases (e.g., malignant diseases, diabetic neuropathy, and Alzheimer’s disease), and the use of certain drugs (e.g., β-blockers, clonidine, naloxone, and anti-inflammatory drugs). In these conditions, melatonin production is reduced, and individuals often have accompanying sleep disorders [11,12].

The most common indication for systemic use of exogenous melatonin is sleep disorders (insomnia). Melatonin synchronizes circadian rhythms and improves sleep onset, duration, and quality [13]. Melatonin can be taken as a supplement, which is well tolerated and has no known short-term or long-term adverse effects [14,20]. Melatonin is classified as a dietary supplement that is not regulated by the Food and Drug Administration and can be purchased in any dose without a doctor’s prescription [14,20]. In Europe, melatonin has been approved for the management of primary insomnia in adults over the age of 55 [14].

There are several ways to administer melatonin, including tablets, oral solutions, sprays for the nose or oral mucosa, and in the form of skin patches, topical creams, or hydrogels [21,22]. The most common side effects of systemic melatonin use are mild and include headache, nausea, dizziness, and drowsiness [21]. However, caution is required in the case of polypharmacy (the simultaneous use of a large number of drugs) since melatonin use can affect the metabolism of drugs that are also metabolized by cytochrome p450 (such as anticoagulants and antithrombotic drugs, anticonvulsants, oral contraceptives, oral hypoglycemics, and immunosuppressants) [21]. According to recent recommendations by an expert group (International Expert Opinions and Recommendations) on the use of melatonin for insomnia, 2–10 mg of slow-release melatonin taken one to two hours before bedtime is recommended [15].

There are also various other indications for melatonin use. For jet lag, a dose of 0.5–1.0 mg of systemic melatonin can be used [16]. Because of its antioxidant and anti-inflammatory properties, melatonin is also used as a natural dietary supplement for athletes for sleep cycle regulation and to protect muscles from oxidative stress [23]. Its effectiveness has also been proven (according to clinical cohort studies) in children with autistic disorders, women with premenstrual dysphoric disorder, hypertensive patients taking beta-blockers, and children with attention deficit hyperactivity disorder (ADHD) [24,25,26,27].

## 4. Melatonin in Skin Diseases

Due to the relative safety of melatonin use, its favorable effect on the sleep cycle, and numerous antioxidant, anti-inflammatory, and antitumor effects, melatonin continues to be studied in patients with chronic and malignant diseases, including various skin conditions [3,17,18,19,28,29,30,31,32,33,34,35,36,37,38,39,40,41,42,43,44,45]. According to the literature data, there are many proven uses for melatonin in dermatologic conditions: Photoprotection (due to melatonin’s antioxidant effects and role in preventing damage due to DNA repair mechanisms), antitumor activity, hypopigmentation processes, maintaining the integrity of the epidermal barrier, healing skin, anti-inflammatory, and immunomodulating activity in inflammatory dermatoses, and the thermoregulation of the body [3]. However, a study of the current use of melatonin in dermatology indicates that it has not yet taken its place in dermatological therapy despite its potential [3]. A search of the literature shows that the systemic application of melatonin is most often used in atopic dermatitis (AD) and melasma (Table 1) [17,18,19,28]. Aside from pineal production of melatonin, the skin and hair follicles produce local melatonin and also possess receptors for melatonin [3]. Thus, they significantly participate in the extrapineal synthesis of melatonin, and the level of melatonin in the skin is much higher than in serum levels [3]. In addition, melatonin in the skin structures also participates in the production of a multifunctional methoxyindole, an important metabolite and factor in the maintenance of skin functions and anti-inflammatory and antioxidant reactions. This helps to prevent pathological changes in the skin by modulating the skin’s response to exogenous agents from the environment [3]. Since melatonin is a molecule with distinct potential in the field of dermatology, results from further studies could lead to wider use of melatonin (in both topical and systemic forms).

In addition to systemic melatonin administration (such as for patients with sleep disorders), therapeutic applications of topical melatonin are also possible for certain dermatological diseases, as well as for some diseases of the oral cavity. Thus, topical melatonin can be used for pigmentation disorders (such as melasma) and in scalp diseases (such as androgenic alopecia and telogen effluvium) (Table 1) [29,30]. According to research, in patients with periodontitis, improvements in key periodontal parameters were seen after topical and systemic melatonin use, such as reductions in pocket depth and clinical loss of attachment [31].

## 5. Melatonin in Allergic Skin Disorders

Concerning allergic skin conditions and melatonin’s role in the skin, it is important to note the skin’s ability to synthesize melatonin itself, which is an important function of the skin in the stress response [46]. Melatonin and its metabolites in the skin protect against various harmful substances from the environment and serve an anti-inflammatory purpose in numerous inflammatory conditions [46]. In cells, melatonin is metabolized in the mitochondria, and it is assumed that melatonin significantly affects cell energy homeostasis and participates in the anti-apoptotic processes and antioxidation responses [46]. It is believed that melatonin and its metabolites could coordinate mitochondrial interactions with skin cells to determine whether it will survive or enter a well-defined differentiation pathway necessary to form the epidermal barrier or die via apoptotic pathways to prevent carcinogenesis [46]. In addition, the skin contains functional melatonin receptors, so melatonin can be a promising candidate for maintaining and protecting the epidermal barrier [47]. Experimental data have already shown that melatonin inhibits the development of atopic dermatitis, as we previously mentioned, so melatonin can also be a potential therapeutic option in protecting skin integrity and helping to maintain a functional epidermal barrier in patients with atopic dermatitis. Thus, melatonin and its metabolites act as a powerful scavenger of free radicals, inhibiting their formation through the activation of the enzyme flavoprotein quinone reductase II in the cell cytosol, which is the basis of melatonin’s antioxidant role in photoprotection and carcinogenesis [47].

According to literature data on the current use of melatonin in dermatology, the systemic application of melatonin is most often used in atopic dermatitis, one of the most common chronic allergic disorders, especially in urban areas [17,18,19]. Among all skin diseases, melatonin’s usefulness has most often been linked to atopic dermatitis [32,33,34,35,36,37] (Table 2). It has been well established that atopic dermatitis is characterized by intense itching and the consequent disturbance of sleep quality, making melatonin a therapeutic option [17,18,19]. The main features of atopic dermatitis are skin dryness and the appearance of eczematous lesions, which are often accompanied by an intense feeling of itching [37]. The disease most often leads to impaired quality of life due to various disease-related features such as itch and reduced self-confidence due to visible skin lesions [48]. In these patients, multiple causes contribute to sleep disturbances as well, including learned scratching behavior and increased monoamines, also leading to impaired quality of life [36].

A search of the literature also indicates that atopic dermatitis is the most frequently studied dermatological disorder concerning melatonin concentrations/values [32,33,34,35,36,37]. The oldest article on this topic, published in 1988, analyzed serum melatonin values (measured every two hours) in 18 patients with severe atopic dermatitis, of which only four patients had normal serum melatonin values [37]. A recent article investigated serum melatonin values in 36 patients with severe and extremely severe forms of atopic dermatitis [33]. It found that patients with very severe atopic dermatitis had significantly lower serum melatonin concentrations when compared to patients with a severe clinical picture [33]. According to a study conducted in children with atopic dermatitis, the administration of oral melatonin led to a reduction in sleep latency by 21.4 min (compared to a placebo), as well as a decrease in disease severity (measured by SCORAD) [19]. The study did not record any side effects of systemic melatonin use or any adverse events, which is particularly important when it comes to the pediatric population [19]. Since this disease is characterized by an intense subjective feeling of itching that disturbs the quality of sleep, the use of systemic melatonin in patients with atopic dermatitis is primarily based on melatonin’s role in improving sleep quality.

Aside from its direct effects on the sleep cycle, it is possible that melatonin could have immunomodulatory and anti-inflammatory effects on allergic diseases [49]. The activation of the immune system leads to increased production of free radicals associated with decreased melatonin levels and depressed antioxidant enzyme activities in inflammatory skin diseases [50]. Moreover, in allergic disorders, such as atopic dermatitis and urticaria, mast cells are activated, which leads to the degranulation of proinflammatory cytokines [49]. Data from experimental studies have shown that melatonin reduces inflammation in atopic dermatitis and reduces total serum IgE and IL-4 [49]. In contrast, in bronchial asthma, an allergic disorder of the respiratory tract, melatonin actually has a pro-inflammatory effect, so its use is not recommended for asthma [51]. Clinical data from experimental studies are still limited.

Recently, for the first time, melatonin levels have been analyzed in patients with urticaria, another allergic disease [52]. Urticaria is a skin disease characterized by hives and angioedema, and often intense itching. It is most often caused by allergic reactions, but it can also be caused by many non-allergic factors and mechanisms [52,53,54,55,56,57,58,59,60]. By duration, urticaria can be classified as acute or chronic. Chronic urticaria is defined by the appearance of urticaria (alone or in combination with angioedema) over a period of time longer than six weeks, accompanied by an intense subjective feeling of itching [52,53]. Depending on the cause or trigger, chronic urticaria can be classified as inducible or spontaneous. The most common causes of inducible urticaria are heat, cold, pressure on the skin, exercise, water, vibration, and sunlight [58]. Among all forms of chronic urticaria, the most common form is chronic spontaneous urticaria (CSU), urticaria without a known cause or trigger. CSU is often accompanied by various comorbidities, including atopic diseases, autoimmune disorders, infections, and psychiatric disorders [54,55,56,57,58,59]. Considering the long duration of the disease, the inability to identify the cause or trigger of the disease, and the resistance to therapeutic options, CSU is one of the dermatological diseases that most greatly affects the quality of life and psychological condition of patients. Therefore, it is not surprising that CSU is one of the dermatological diseases with the largest number of psychiatric comorbidities and that sleep disorders are the most common psychologic/psychiatric disorders in CSU patients [59,60]. Considering these data, melatonin supplementation as a possible new therapeutic option has been considered. A recently conducted pilot study on melatonin values in patients with CSU showed low melatonin values in these patients’ saliva [52]. These initial results need to be supported by further studies with greater numbers of patients, including more experimental research on the possibility of using melatonin supplements in CSU patients.

## 6. Melatonin in Non-Allergic Skin Disorders

Among other non-allergic skin diseases, in which melatonin use has been examined, is melasma, a pigmentation disorder that occurs most often in dark-skinned women and is very often associated with hormonal imbalances and other disorders [28,29]. The disease is frequently resistant to therapy and prone to relapses. In 2010, a study was conducted on 36 patients with melasma who received melatonin orally (3 mg) together with topical melatonin applied to hyperpigmented areas of skin for three months; 10 patients were given a placebo preparation [28]. The combined application of melatonin yielded a significant improvement in the disease [28], which points to the antioxidant effects and hypopigmentation effects of melatonin on hyperpigmentation disorders [29]. Topical melatonin is also used for androgenetic alopecia [30] due to melatonin’s direct and indirect effects on hair follicles (indirectly through the regulation of other hormones that improve hair growth) [3]. Interestingly, in 2021, a study was published that linked rosacea and Alzheimer’s disease [38]. In establishing a common molecular regulatory network and identifying potential therapeutic drugs for rosacea and Alzheimer’s disease, melatonin was among the 113 predicted drugs for these diseases [38]. Administration of melatonin significantly improved rosacea-like skin changes, most likely through its anti-inflammatory and angiogenesis effects [38]. In another study, serum melatonin was measured in patients with rosacea in perimenopausal women (15 women with rosacea and 15 healthy women of similar age) [39]. In the patients with rosacea, decreased serum melatonin values were observed. The drop in concentration correlated with the severity of menopausal syndrome [39].

Serum melatonin has also been investigated in psoriasis and vitiligo [40,41]. In a study conducted in 1988, serum melatonin values were analyzed in 13 patients with psoriasis in comparison to 13 healthy subjects [40]. According to the results, in the serum of patients with psoriasis, significantly lower melatonin values were observed, and they lost the nocturnal increase in melatonin concentrations and the usual circadian rhythm of melatonin secretion [40]. Serum melatonin was measured in a study on 41 patients with vitiligo (16 patients with the segmental form and 25 with the non-segmental form) who were exposed to stress and who had a history of high levels of experienced stress [41]. According to the results, among patients with segmental vitiligo, a reduced melatonin concentration was seen in 8 out of 16 patients, while among patients with non-segmental vitiligo, reduced melatonin levels were recorded in all 25 patients [41].

In addition to its therapeutic effects, melatonin plays a role in preventing the appearance of pathological skin changes that are the result of skin aging. The main role of melatonin in the skin is to reduce cell apoptosis, modulate the expression of melatonin (MT2) receptors, and reduce the expression of the alpha-estrogen receptor [42,43]. It also has an antioxidant effect, which contributes to the fight against free radicals and stimulates the repair of damaged skin and cellular DNA [53]. The most significant factor in the etiology of skin aging is exposure to UV radiation (UVR), which is also a crucial environmental factor in the development of skin tumors [42,43]. It has been shown that topical melatonin reduces the formation of free radicals and supports photoprotection measures (the use of creams with a high protection factor or wearing protective clothing and glasses). Since melatonin has been shown to be safe and effective, topical melatonin itself can also be used as a photoprotection measure [42,43].

There is considerable evidence that melatonin is an effective anti-aging compound for the skin—a crucial skin protectant, from free radical scavenging to DNA damage repair. Melatonin penetrates the skin and activates the synthesis of antioxidant enzymes, which block free radicals and repair oxidative damage to the skin. Moreover, regarding UV solar damage, melatonin distinctly counteracts the massive generation of reactive oxygen species and mitochondrial and DNA damage; melatonin ameliorates (external factors-induced) DNA damage and has an antiapoptotic effect [42,44]. At the mitochondrial level, melatonin directly scavenges ROS and inhibits m-iNOS expression, which, in mitochondria, neutralizes both reactive oxygen and nitrogen species (ROS/RNS), which improves oxidative phosphorylation and ATP production. Moreover, skin aging is often associated with the absence of estrogen during menopause. A lack of estrogen in the skin is associated with decreased epidermal and dermal thickness, wrinkling, dryness, atrophy, decreased collagen, elasticity loss, and poor hair growth; however, according to previous research on postmenopausal rats, melatonin increased epidermal thickness, the subcutaneous fat layer, and elastic fibers (when compared to controls) [44]. In another study, following melatonin treatment on animal models, an increase in fibroblast growth factor-β (FGF-β), collagen type I, and fibronectin, as well as high c-Myc and β-catenin expression of the epidermis and hair bulb, were seen [45]. It should also be mentioned that melatonin may penetrate the skin and accumulate in the stratum corneum through prolonged release from the skin into the bloodstream (over a 24 h period) [44].

In addition, a recent case report looked at the topical application of melatonin cream for the treatment of acute radiodermatitis in a breast adenocarcinoma patient [46]. Melatonin cream use for three weeks led to the regression of radiodermatitis, which points to another possible therapeutic option for acute radiodermatitis in addition to topical corticosteroid preparations and emollient creams [46]. The use of melatonin in the treatment of melanoma is particularly interesting since melanoma is the most malignant skin tumor, and its incidence is only increasing. Namely, due to melatonin’s oncostatic potential and antitumor properties, as well as the abovementioned antioxidant, anti-inflammatory, and apoptosis induction effects, in some studies, melatonin was included in the adjuvant therapy of melanoma together with chemotherapy [47]. It has been shown that melatonin, as a synergistic molecule, can improve the outcome of these patients [47].

According to literature data, melatonin effectively acts against UVR (prominent inducers of epidermal damage), skin cancer, and DNA photodamage. One of the common UVR-induced stress proteins is heat shock protein 70 (Hsp70), highly expressed in human keratinocytes, providing cellular resistance to such stressors. According to research on the interaction of melatonin and Hsp70 toward UVR-induced inflammatory/apoptotic responses (using human full-thickness skin and normal human epidermal keratinocytes), UVR upregulated Hsp70 gene expression in human epidermis, while melatonin significantly inverted this effect. Similar regulation patterns were seen in Hsp70 protein levels, and studies involving silencing of Hsp70 RNA (Hsp70 siRNA) showed decreased IκB-α (an inhibitor of NF-κB) and enhanced gene expression of pro-inflammatory cytokines (IL-1β, IL-6, Casp-1) and pro-apoptotic protein (Casp-3) in NHEK [48,49].

A search of the current literature shows an increase in melatonin research, not only in dermatology but also in other areas of medicine, such as psychiatry, oncology, neurology, pediatrics, endocrinology, and gynecology.

## 7. Role of Melatonin in Other Allergic Diseases Potentially Related to Skin Allergic Diseases

In allergic diseases, melatonin has an important role in their neuroimmunological and immunomodulatory processes [61]. However, its potential use in patients with allergic diseases has rarely been considered. Previous studies of patients with allergic asthma have indicated that melatonin, regulating smooth muscle tone and influencing the immune response, could be involved in the regulation of bronchial hyperreactivity [62]. Therefore, in asthma, melatonin may act as a pro-inflammatory agent leading to bronchial constriction [62]. It is known that allergy and autoimmunity are considered two potential outcomes of a dysregulated immune system. Furthermore, a strong positive association between Th2-mediated allergic disorders and Th1-mediated autoimmune disorders has been recorded by some studies.

Atopy is an individual/family tendency to produce IgE antibodies in response to low doses of allergens and to develop manifestations of allergic diseases such as atopic dermatitis, allergic rhinitis, and asthma [61]. Different atopic/allergic diseases very often occur in the same patient, i.e., they are often associated. Thus, atopic dermatitis in childhood often progresses to other types of allergic diseases such as asthma and allergic rhinitis. This process is called the atopic march (a combination of atopic dermatitis, allergic rhinitis and allergic asthma), although the specific explanation of how asthma arises from atopic dermatitis is still not fully understood [63]. Oxidative stress is one of the key pathogenetic mechanisms in the development of allergic asthma. According to previous studies on allergic asthma, there is a potential link between increased oxidative damage and the occurrence/exacerbation of allergic asthma, with thymic stromal lymphopoeitin (TSLP) playing the key role in the atopic march, by stimulating sensitization to allergens in the skin with a damaged barrier [63]. According to results from research on an experimental atopic march mouse model, in lung cells, oxidative stress significantly activates NF-κB signaling pathways, which cause TSLP release, again illustrating melatonin’s antioxidant properties [62]. According to literature data, melatonin treatment, in addition to reducing the production of free oxygen radicals, also reduces the level of lipid peroxidation [63]. From a therapeutic point of view, in the lungs, antioxidants such as melatonin can inhibit the activation of NF-κB, especially after the onset of atopic dermatitis [63]. In the pathogenesis of allergic diseases, Th2 and Th1 cells are very important pathogenetic factors. Typically, CD4 + T cells are transformed into Th1 cells (producing IFN-γ and IL-2) and Th2 cells (producing IL-4, IL-9, IL-10, and IL-13), which enable eosinophils’ maturation and activation [2]. Thus, in allergic rhinitis, this pathway shifts to a Th2 increase (Th1 and IFN-γ levels decrease). As is known, IFN-γ is released primarily from Th1, cytotoxic T, naïve CD4 + T, and NK cells, and secondarily from dendritic cells, macrophages, and even B cells [2]. Genetic factors must also be mentioned here. IFN-γ regulates approximately 500 genes, including increased transcription of antigen-presenting cells’ genes, differentiation of naïve CD4 + T cells towards Th1 cell direction, increased apoptosis and B cell response, increased immunoglobulin heavy chain production (increased IgG and decreased IgE), and the regulation of various inflammatory cells cytokines and adhesion molecules, also important for atopic dermatitis [2]. IFN-γ suppresses the Th2 response and increases the Th1 response, reduces eotaxin receptor expression, inhibits eosinophil differentiation, and regulates the eosinophilic response [2]. IFN-γ also increases nitric oxide (NO) synthesis by inducing the NO synthase enzyme, preventing IgE-mediated mast cells’ degranulation [2]. This relationship between melatonin and IFN-γ levels was recorded in patients with allergic rhinitis, who had lower serum IFN-γ levels than controls [2]. Low melatonin levels reduce IFN-γ synthesis, which reduces melatonin synthesis, while IFN-γ stimulation increases melatonin synthesis [2].

According to the results of an experimental study on mice with the atopic march, the excessive development of oxidative stress in the lungs significantly activated the signaling pathway (NF-κB), which caused the further release of pro-inflammatory cytokines and the development of an inflammatory reaction in atopic diseases [63]. The use of antioxidants, such as melatonin in atopic disorders, can also be directed to reduce the activation of NF-κB in the lungs, especially after the onset of atopic dermatitis [63]. So, the role of oxidative stress is very important for allergic disorders. An excessive number of free oxygen radicals and other oxidation products can lead to extended oxidative damage, seen in the pathogenesis of allergic asthma and atopic dermatitis. It is also notable that lung cells (particularly alveolar epithelial cells type 2) are specifically sensitive to the effect of free oxygen radicals and the secretion of pro-inflammatory cytokines (TNF-α and IL-1β) [63]. Moreover, during the oxidation process, transcription factors such as NF-κB, activator protein-1 (AP-1) and hypoxia-inducible factor (HIF)-1 are activated [63]. It is important to mention that TSLP, a cytokine derived from the airway epithelium, plays a central role in the polarization of dendritic cells and promotes the differentiation of naïve T cells into Th2 cells [63]. Lung and bronchial fibroblasts, as well as smooth muscle cells, can also produce TSLP in inflammatory conditions that are promoted by IL-13 stimulation [63]. In addition, studies have shown that keratinocyte TSLP plays a key role in the development of atopic dermatitis into asthma. In allergic asthma, TSLP activates dendritic cells and promotes a shift from Th1/Th2 homeostasis to a Th2 response, leading to airway remodeling and persistent hyperreactivity [63].

Moreover, autoimmune thyroid diseases are the most common of all autoimmune pathological conditions potentially affected by melatonin, The possible association between the function/role of melatonin and the thyroid gland was first reported by Kvetnoy, who proved that melatonin is synthesized by thyroid neuroendocrine C cells and that melatonin receptors (MT)1 are present in follicular cells [61]. Furthermore, several studies have also suggested a paracrine role for melatonin in the regulation of thyroid activity, indicating that the use of melatonin in conditions with increased oxidative stress could be useful for reducing oxidative processes involved in autoimmune thyroid diseases [61].

In addition, literature data show impressive neuroimmunological effects of melatonin, including neuroprotective and antiexcitatory action—it inhibits the formation of free radicals and modulates the antioxidant actions of vitamins in neurons—and the induction of the pattern of regulatory T cells and immunomodulatory cytokines. It also influences certain conditions (e.g., in rheumatoid arthritis, it increases inflammatory mediators and can aggravate clinical manifestations) [64].

## 8. Additional Considerations concerning the Use of Melatonin in Dermatologic Allergic and Non-Allergic Conditions

Currently, the available results of previous studies on melatonin use in dermatological conditions indicate the predominantly useful effects of melatonin on different skin conditions; thus, the future of melatonin use for skin diseases looks promising. However, only a limited number of diseases has been examined so far. In order to obtain more data on these useful effects of melatonin for allergic and non-allergic dermatologic conditions, more studies (with greater numbers of participants) need to be conducted to analyze skin conditions/diagnoses.

When reviewing the research literature, the positive effects of melatonin are primarily mentioned. In some cases, melatonin has demonstrated limited effectiveness, and very rarely, it has demonstrated harmful effects. Among the side effects of melatonin use, the most common are headaches, nausea, and dizziness [65]. Therefore, caution is required when using oral melatonin in the elderly population, because oral melatonin may decrease blood pressure and cause hypothermia [63]. In 2016, the National Agency for the Safety of Medicines and Health Products from France published a list of 200 side effects associated with the use of melatonin (reported between 1985 and 2016): 43% were neurological disorders (such as convulsion, syncope, and headache), 24% were psychiatric disorders (anxiety and depression), 19% were skin disorders (rashes and maculopapular rashes), and 19% were digestive problems (constipation, nausea, and acute pancreatitis) [65].

Finally, similarities in the usefulness of melatonin and vitamin D have recently been examined. Both act as hormones, affect multiple body systems through their immune-modulating and anti-inflammatory functions, and they are also both synthesized in the skin depending on exposure to light or darkness [66]. For all its possible benefits, melatonin has become a popular dietary supplement, predominantly for the improvement of sleep quality. Likewise, recent research on melatonin in numerous inflammatory and allergic conditions, as well as malignant diseases, has led to greater awareness of melatonin’s ability to act as a powerful antioxidant and immuno-active agent [66].

## 9. Conclusions

Melatonin is a molecule with many useful roles for organisms that can also be used for therapeutic purposes. Considering the many therapeutic possibilities, there is also great potential for the use of melatonin in dermatology, a field in which it is currently rarely used. One of the most important roles of melatonin is the initiation and maintenance of the sleep cycle, an important fact to consider in clinical approaches to chronic allergic skin diseases that are accompanied by sleep disorders, such as atopic dermatitis and CSU. Due to its antioxidant, anti-inflammatory, and hypopigmentation effects, it can also be used for other inflammatory skin conditions and hyperpigmentation disorders, and also as a photoprotection measure.

The research results and trends presented in this paper and the above discussion of melatonin’s effects on the body indicate that melatonin is a molecule with distinct potential for therapeutic uses in the field of dermatology. New research in this area will enable the wider use of melatonin in clinical practice in the future.

## Figures and Tables

**Table 1 ijms-24-04039-t001:** Possible effects of melatonin in skin disorders.

Disorder		Effects of Melatonin
**ALLERGIC** **DISORDERS**	**Atopic dermatitis**	- Antiinflammatory - Immunomodulatory - Antioxidant - Sleep promoting
	**Chronic urticaria**	- Antiinflammatory - Imunomodulatory - Sleep promoting
**NON-ALLERGIC DISORDERS**	**Melasma**	- Hypopigmentary - Antioxidant
	**Androgenetic alopecia** **Telogen effluvium**	- Improving hair growth - Antiinflammatory
	**Rosacea**	- Antiinflammatory - Immunomodulatory

**Table 2 ijms-24-04039-t002:** Reports on melatonin in allergic disorders.

Authors/Year of Publication	Research	Type	Subjects	Results
Dwiyana et al., 2022 [32]	Melatonin and sleep disturbances in children with atopic dermatitis	Experimental study	19 children with moderate AD and 19 participants without AD	Children with moderate AD have impaired sleep quality and lower melatonin levels compared to controls.
Jaworek et al., 2021 [33]	Melatonin and sleep disorders in patients with severe atopic dermatitis	Experimental study	36 adult patients with severe and very severe AD and 20 healthy subjects	Significantly lower melatonin concentration in patients with very severe AD (*p* < 0.001) than in patients with severe AD.
Devadasan et al., 2020 [34]	Role of serum melatonin and oxidative stress in childhood atopic dermatitis: a prospective study	Experimental study	30 patients with AD, aged 6 months to 12 years, and an equal number of age and sex-matched controls	The serum levels of malondialdehyde and melatonin were significantly higher among subjects compared to controls.
Shi & Lio, 2019 [17]	Alternative treatments for atopic dermatitis: an update	Review	A literature search was conducted in the PubMed, EMBASE, Cochrane Central Register of Controlled Trials, and Global Resource for EczemA Trials (GREAT) databases for RCTs on complementary and alternative therapies in AD from March 2015 through May 2018, 15 studies	The preliminary results (for many treatments such as vitamin E, East Indian Sandalwood Oil, melatonin, L-histidine, and Manuka honey) show positive clinical effects, but without enough evidence currently to recommend their use for AD therapy.
Patel et al., 2018 [18]	Managing sleep disturbances in children with atopic dermatitis.	Review		When patients do not respond to first-generation antihistamines, melatonin use is recommended before bedtime
Chang et al., 2016 [19]	Melatonin supplementation for children with atopic dermatitis and sleep disturbance: a randomized clinical trial	Experimental study	73 children and adolescents (aged 1 to 18 years) with physician-diagnosed AD (involving at least 5% of the total body surface area). 48 children were randomized 1:1 to melatonin or placebo treatment, and 38 of these (79%) completed the cross-over period of the trial.	After melatonin treatment of 48 children, the SCORAD index decreased (by 9.1 compared with the placebo), from a mean (SD) of 49.1 (24.3) to 40.2 (20.9). Also, the sleep-onset latency shortened by 21.4 min (to those given the placebo).
Chang et al., 2014 [35]	Atopic dermatitis, melatonin, and sleep disturbance	Experimental study	72 patients with AD and 32 controls ages 1 to 18 years.	Lower nocturnal melatonin secretion was significantly associated with sleep disturbance in AD patients
Kimata, 2007 [36]	Elevation of salivary melatonin levels by viewing a humorous film in patients with atopic eczema	Experimental study	12 patients with AD and 12 healthy controls	Reduces salivary melatonin levels in AD patients compared to healthy controls
Schwarz et al., 1988 [37]	Alterations of melatonin secretion in atopic eczema	Experimental study	18 AD patients and 40 healthy controls	Low serum melatonin levels were determined in 6 patients, increased serum melatonin levels in 8 AD patients and only 4 patients had normal serum melatonin levels compared with the controls (*n* = 40).

Atopic dermatitis = AD.

## Data Availability

Main resources for this research were PubMed and PMC.

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
