# Peer review of "Melatonin in Dermatologic Allergic Diseases and Other Skin Conditions: Current Trends and Reports"

_ijms, 2023, doi:10.3390/ijms24044039_

Round 1

Reviewer 1 Report (Previous Reviewer 2)

Dear Author,

the paper entitled "Melatonin in dermatologic allergic diseases and other skin conditions: current trends and reports" is well structured and easy to understand. 

Just one point: table 1 is poor in content it would not seem necessary that information can be put in the text.

Author Response

Dear Reviewer, thank You for Your recommendations to improve this manuscript. According to Your suggestions, we expanded Table 1. with melatonin effects in specific disorder.

To improve our manuscript, we have been added several paragraphs in the manuscript about the melatonin activation via receptors, the effects of melatonin in allergic skin diseases, as well as, the possible side effects of melatonin use at the end of the manuscript.

We hope we have improved our manuscript with this addition.

Nice regards, Authors

Reviewer 2 Report (New Reviewer)

The authors discussed the potential role of melatonin in skin diseases suggesting that this compound may be of help in the treatment. However, the current data supporting this idea are at most weak. The authors are too optimistic and try to convince the readers to use this compound, without pointing out the weakness of currently available data. In my opinion, this review is not suitable for a highly-ranked journal. The authors should be more critical while discussing the previous results. Otherwise, this is pure speculation, that e.g. melatonin can help in the treatment of AD patients. 

Author Response

Dear Reviewer, thank You for Your opinion. To improve our manuscript, we have been added several paragraphs in the manuscript about the melatonin activation via receptors, the effects of melatonin in allergic skin diseases, as well as, the possible side effects of melatonin use at the end of the manuscript. Also, we expanded Table 1. with melatonin effects in specific disorder.

We hope we have improved our manuscript with this addition.

Nice regards, Authors

Reviewer 3 Report (New Reviewer)

The authors described that the effect of melatonin on skin.

It is well-known that melatonin affects on sleep cycles.

But the authors adequately commented of other aspect of melanin, and I think this paper would prompt future studies, for example relationship of skin disorders and melatonin.

Taken together, I think this manuscript is acceptable after minor revision.

Comments;

l85 MTR1, MTR2  describe the full name of receptor, and it is preferable to explain the signal transduction caused by melatonin.

l93 "between 00:00h and 05:00h" Does it mean after fallen asleep?

Author Response

Dear Reviewer, thank You for Your recommendations to improve this manuscript.

According to Your suggestions, to improve our manuscript, we have been added several paragraphs in the manuscript about the melatonin activation via receptors where we explained the signal transduction.  We also, have been written the new paragraph on the effects of melatonin in allergic skin diseases, as well as, the possible side effects of melatonin use at the end of the manuscript. Also, we expanded Table 1. with melatonin effects in specific disorder.

Considering I93, it is correct. It means that the highest melatonin concentrations are reached between 00:00h and 05:00h, but the patient had to be awake to provide a blood, saliva or urine sample for analysis.

We hope we have improved our manuscript with this addition.

Nice regards, Authors

Round 2

Reviewer 2 Report (New Reviewer)

The manuscript has been improved. However, the authors still should temper some of the statements, as currently, there are no good studies confirming such statements, e.g.:

- there are many proven uses for melatonin in dermatologic conditions

- In addition to its therapeutic effects, melatonin plays a role in preventing the appearance of pathological skin changes that are the result of skin aging. 

- In allergic diseases, melatonin has an important role in their neuroimmunological and immunomodulatory processes

Author Response

Dear Reviewer,

Thank You for Your opinion.

According to Your suggestions,

We have done professional language corrections.

Also, we added suggested statements and paragraphs about preventive role of melatonin in skin aging. Please, see the attachment... 

We are available for all inquiries. Thank You and kind regards,

Authors

This manuscript is a resubmission of an earlier submission. The following is a list of the peer review reports and author responses from that submission.

Round 1

Reviewer 1 Report

The manuscript deals with the melatonin role in dermatologic allergic diseases and other skin conditions. When I was reading the article title I was expected to read about melatonin's role in improving skin conditions. I found this expectation met only in the section "Melatonin in non-allergic skin conditions". All the other sections highlighted the melatonin (secondary) role in improving the sleep quality in skin conditions in which sleep disorders are involved.

Please find here below some recommendations:

-          “The melatonin molecule” section is quite long. It describes in detail the molecule synthesis, bioavailability, and its role in sleep regulation. In relation to the melatonin role in dermatologic allergic diseases and other skin conditions (the main topic of the review) I would delete or at least I would consider reducing the length of the description of the melatonin role for sleep regulation (lines from 100 to 126).

-          Reference 19 should be substituted with the appropriate link to the EFSA opinion (e.g., the opinion reported in this link https://www.efsa.europa.eu/it/efsajournal/pub/1467). Related to “… over 55 years of age [19]” I did not find in the EFSA document this indication.

-          The “Melatonin as a therapeutic option” same consideration of the “The melatonin molecule section”. This section is quite long in relation to the main topic of the review. I would recommend reducing the length of this section.

-          Line 171 “… tions [2] In the literature data …” please add the punctuation point.

-          The “Melatonin in skin diseases” section reports different proven uses for melatonin in dermatologic conditions, however, these different uses are supported by only one reference [Ref. 2]. Please add to each recommended use indication the appropriate and updated reference.

-          Line 205-222 please consider shortening the AD description paragraph.

-          Line 240-267 please consider shortening the AD description paragraph. 

Author Response

Dear Reviewer, thank You for Your recommendations to improve this manuscript. According to Your suggestions, we:

  1. Shortened “The melatonin molecule” section

Melatonin is an important physiological sleep regulator and its adequate production is very important to achieve a good sleep quality. Factors that have a negative effect on melatonin production are aging, the presence of certain diseases (e.g., malignant diseases, diabetic neuropathy, Alzheimer's disease) and the use of certain drugs (e.g., β-blockers, clonidine, naloxone and anti-inflammatory drugs). In these conditions, melatonin production is reduced, and individuals often have accompanying sleep disorders [10].  

The sleep-wake cycle is the most well-known activity of the human body that follows the circadian rhythm [10]. Other activities in the body that also follow the circadian rhythm are changes in body temperature, blood pressure, the immune response, and changes in the level of certain hormones in the body [10].  Sleep is a neurochemical process important for restoring energy to the brain by turning off the flow of external information and processing information acquired during the waking state [10].  Sleep enables cerebral changes that allow for learning, memory and the activation of the glymphatic system, responsible for cleaning metabolites in the brain [10]. These functions are essential for brain development, a person's physical and mental health, and the maintenance of cognitive functions that result in a sense of well-being [10]. Long-term sleep problems may negatively influence overall health by weakening the immune system and increasing the risk of hypertension, cardiovascular disease, and insulin resistance [11]. The most common sleep-related disorders include difficulty falling asleep, early waking, and a feeling of fatigue that disrupts daily activities and consequently leads to difficulties in the individual's work and social life [11]. Because melatonin is an important physiological sleep regulator, its adequate production is very important. Factors that have a negative effect on melatonin production are aging, the presence of certain diseases (e.g., malignant diseases, diabetic neuropathy, Alzheimer's disease) and the use of certain drugs (e.g., β-blockers, clonidine, naloxone and anti-inflammatory drugs). In these conditions, melatonin production is reduced, and individuals often have accompanying sleep disorders [10].  

The positive effect that melatonin has on sleep makes it useful for treating numerous diseases and disorders. In most patients, it improves sleep—melatonin dampens the phase of the circadian rhythm that promotes wakefulness, thereby promoting sleep [12]. Therefore, several studies showed that exogenous intake of melatonin in the diseases accompanied with sleep disorders increases body concentration of melatonin and favorably affects quality of sleep, so the systemic application of melatonin in sleep disorders was generally accepted [12-18].

  1. Excluded reference about EFSA opinion and added a new reference [19] which was suggested by the Reviewer #2.

The most common indication for systemic use of exogenous melatonin is sleep disorders (insomnia). Melatonin synchronizes circadian rhythms and improves sleep onset, duration and quality [12]. Melatonin can be taken as a supplement, which are well tolerated and have no known short-term or long-term adverse effects [13,19]. Melatonin is classified as a dietary supplement which is not regulated by the Food and Drug Administration and can be purchased in any dose without a doctor's prescription [13,19]. In Europe, melatonin has been approved for the management of primary insomnia in adults over the age of 55 [13].

Notably, melatonin products are registered differently in the US and EU.  In the US, according to the FDA ("Food and Drug Administration"), melatonin is considered a dietary supplement and can be purchased in any dose without a doctor's prescription. In the European Union, according to the EFSA ("European Food Safety Authority"), melatonin is also registered as a dietary supplement, but it is indicated for the treatment of primary insomnia in adults over 55 years of age [19]. According to research data, the use of systemic melatonin and its agonists in patients with insomnia prolongs sleep time, improves sleep quality, and reduces sleep latency [19].  They work by indiscriminately binding to membrane MT1 and MT2 receptors, which activates them [19]. 

  1. Shortened the “Melatonin as a therapeutic option” section

There are several ways to administer melatonin, including tablets, oral solutions, sprays for the nose or oral mucosa, and in the form of skin patches, topical creams or hydrogels [20, 21].  Melatonin is metabolized in the liver by cytochrome p450, and the rate of absorption and metabolism may vary depending on age and gender [19]. According to research results, in younger adult patients with insomnia, serum melatonin concentrations reach their maximum at approximately 45 minutes after ingestion. Also, it is recommended to use melatonin in the evening in order to imitate the physiological production of melatonin, which is based on the circadian rhythm [19].

The most common side effects of systemic melatonin use are mild and include: headache, nausea, dizziness, and drowsiness [20]. However, caution is required in the case of polypharmacy (the simultaneous use of a large number of drugs) since melatonin use can affect the metabolism of drugs that are also metabolized by cytochrome p450 (such as anticoagulants and antithrombotic drugs, anticonvulsants, oral contraceptives, oral hypoglycemics and immunosuppressants) [20]. According to recent recommendations by an expert group (International Expert Opinions and Recommendations) on the use of melatonin for insomnia, 2-10 mg of slow-release melatonin taken one to two hours before bedtime is recommended [14]. 

There are also various other indications for melatonin use. For jet lag dose of 0.5-1.0 mg of systemic melatonin can be used [15]. Because of its antioxidant and anti-inflammatory effects, melatonin is also used as a natural dietary supplement for athletes for sleep cycle regulation and to protect muscles from oxidative stress [22]. Recent research shows that intense exercise disrupts athletes’ antioxidant status, and melatonin supplementation strengthens it [20]. Its effectiveness has also been proven (according to clinical cohort studies) in children with autistic disorders, women with premenstrual dysphoric disorder, hypertensive patients taking beta-blockers, and in children with attention deficit hyperactivity disorder (ADHD) [23-26].

  1. Added the punctuation point in the line 171

… including various skin conditions [2,16-18,27-44]. In the literature data,…

  1. Added the appropriate and updated reference for each indication that were mentioned in our manuscript.

… including various skin conditions [2,16-18,27-44].  

  1. Shortened 205-222 paragraph

AD is characterized by intense itching and the consequent disturbance of sleep quality, making melatonin a therapeutic option for patients [16-18]. Crucial factors in atopic dermatitis development are gene mutation of filaggrin and familial occurrence of atopic diseases. Other risk factors include living in an urban environment, having a higher socioeconomic status and being female [36].  Common triggers for exacerbation of AD are viral infections, food allergens, cosmetics, perfumes, extreme heat and cold, wool, exposure to dust, pollen, mold, cigarette smoke, animal hair, emotional stress, presence of the Staphylococcus aureus bacteria on the skin, among others [36]. Also, changes in the skin microbiome itself are also gaining increasing importance in the understanding of the pathogenesis of AD [36,37]. By frequency, AD occurs in about 20% of the younger population and in about 1-3% of adults [36]. It is also the most common skin disease in children, and 70% of patients with atopic dermatitis develop the disease by the age of 5. The main features of AD are skin dryness and the appearance of eczematous lesions, which are often accompanied by an intense feeling of itching [36]. The disease most often leads to an impaired quality of life due to various disease-related features such as itch and reduced self-confidence due to visible skin lesions [45]. In these patients, multiple causes contribute to sleep disturbances as well, including learned scratching behavior and increased monoamines, also leading to impaired quality of life [35].

  1. Shortened 240-267 paragraph

Among allergic diseases other than AD, melatonin levels were analyzed in patients with urticaria recently for the first time [48]. Urticaria is a skin disease characterized by hives and angioedema, which are often accompanied by an intense subjective feeling of itching. It occurs in about 1% of the world's population, most often in adulthood [39]. The etiopathogenesis of urticaria is complex, and the etiological and provoking factors are numerous, which often leads to therapeutic difficulties in clinical work. The prevalence of any type of urticaria during life is about 20% [39].  Urticaria is most often caused by allergic reactions, but it can also be caused by many non-allergic factors and mechanisms [49-56]. By duration, urticaria can be classified as acute or chronic. Acute urticaria is defined by the appearance of hives (alone or in combination with angioedema) in a period of less than six weeks, accompanied by an intense subjective feeling of itching [36].   It is the most common form of urticaria, which usually regresses after a few days of appropriate therapy. The causes of acute urticaria have been determined in only 50% of patients, and among the most common are drugs, reactions to food and infections of the upper respiratory tract [41].

Chronic urticaria is defined by the appearance of urticaria (alone or in combination with angioedema) in a period of time longer than six weeks, accompanied by an intense subjective feeling of itching [49,50]. Depending on the cause or trigger, chronic urticaria can be classified as inducible or spontaneous. The most common causes of inducible urticaria are heat, cold, pressure on the skin, exercise, water, vibration, and sunlight [54]. Among all forms of chronic urticaria, the most common form is chronic spontaneous urticaria (CSU), urticaria without a known cause or trigger. The prevalence of CSU in the general population is about 1-2%, and it can occur in both children and adults, but it occurs more often in adults [43].  By gender, CSU occurs more often in women, in whom it is twice as common as in men. According to age, the onset of symptoms is most common between the ages of 20 and 40 [43].  The average duration of CSU is two to five years, but in 20% of patients it lasts longer than five years [43].

  1. Additionally, to improve our manuscript, in the paragraph Melatonin in skin diseases, we add a table (Table 1.) where we summarized the most common skin disorders in which melatonin can be used. Also, we added a paragraph about supplementary effects of melatonin in allergic diseases in the paragraph Melatonin in allergic skin diseases. Consequently, an old table 1. became table 2.

Table 1. Possible use of melatonin in skin disorders 

Allergic disorders                                      Non-allergic disorders

Atopic dermatitis                                                Melasma

Chronic spontaneous urticaria                          Androgenic alopecia

                                                                          Telogen effluvium

                                                                          Psoriasis

                                                                          Vitiligo

                                                                          Rosacea

                                                                          Acute radiodermatitis

Besides direct effects on sleep cycle, there is possibility that melatonin has also immunomodulatory and anti-inflammatory effects in allergic diseases [46,47]. The activation of the immune system leads to increased production of free radicals associated with decreased melatonin levels and depressed antioxidant enzyme activities in inflammatory skin diseases [47]. As well, in allergic disorders, such as atopic dermatitis and urticaria, mast cells are activated which leads to degranulation of proinflammatory cytokines [47]. Data from experimental studies showed that melatonin reduces inflammation in AD and reduces serum total IgE and IL-4 [47]. But contradictory, in bronchial asthma, allergic disorder of respiratory tract, has pro-inflammatory effect, so its use is not recommended in asthma [47]. Clinical data from experimental studies are still limited.

We are at your disposal for further inquiries and changes to improve quality of our manuscript.

Kind regards, authors

Reviewer 2 Report

Dear Authors,

the manuscript entitled: " Melatonin in dermatologic allergic diseases and other skin conditions: current trends and reports " shows the possible use of melatonin in the treatment of atopic dermatitis and other skin diseases.

The paper is clear and well written, it present a medical-clinical rather than pharmaceutical-technological tendency.

Some remarks concerning aspects strictly related to the drug and its delivery:

Line 134: Depending on the dosage quantity (≤ 1mg, 1-2mg,> 2mg) the preparations containing melatonin require or not a medical prescription.

The bibliographic reference [19] reported several times in the text could be implemented with the more traditional scientific literature referred to, e.g. Brewster, Glenna S, Barbara Riegel, and Philip R Gehrman “Insomnia in the Older Adult.” Sleep Medicine Clinics 13.1 (2018): 13–19.

Line 142 skin patches or topical creams, I would also add hydrogels as useful pharmaceutical forms by putting some citations proper to hydrogels

Soriano, J.L.; Calpena, A.C.; Rodríguez-Lagunas, M.J.; Domènech, Ò.; Bozal-de Febrer, N.; Garduño-Ramírez, M.L.; Clares, B. Endogenous Antioxidant Cocktail Loaded Hydrogel for Topical Wound Healing of Burns. Pharmaceutics 2021, 13, 8.

Villa, C.; Russo, E. Hydrogels in Hand Sanitizers. Materials 2021, 14, 1577.

Russo, E.; Villa, C. Poloxamer Hydrogels for Biomedical Applications. Pharmaceutics 2019, 11, 671; doi: 10.3390/pharmaceutics11120671

Author Response

Dear Reviewer, thank You for Your recommendations to improve this manuscript. According to Your suggestions, we:

  1. line 134 - changed the whole paragraph about recommendations on the use of melatonin.

 The most common indication for systemic use of exogenous melatonin is sleep disorders (insomnia). Melatonin synchronizes circadian rhythms and improves sleep onset, duration and quality [12]. Melatonin can be taken as a supplement, which are well tolerated and have no known short-term or long-term adverse effects [13,19]. Melatonin is classified as a dietary supplement which is not regulated by the Food and Drug Administration and can be purchased in any dose without a doctor's prescription [13,19]. In Europe, melatonin has been approved for the management of primary insomnia in adults over the age of 55 [13].

Notably, melatonin products are registered differently in the US and EU.  In the US, according to the FDA ("Food and Drug Administration"), melatonin is considered a dietary supplement and can be purchased in any dose without a doctor's prescription. In the European Union, according to the EFSA ("European Food Safety Authority"), melatonin is also registered as a dietary supplement, but it is indicated for the treatment of primary insomnia in adults over 55 years of age [19]. According to research data, the use of systemic melatonin and its agonists in patients with insomnia prolongs sleep time, improves sleep quality, and reduces sleep latency [19].  They work by indiscriminately binding to membrane MT1 and MT2 receptors, which activates them [19]. 

  1. Added Your suggested reference 19
  2. Brewster G.S.; Riegel B.; Gehrman P.R. Insomnia in the Older Adult. Sleep Med Clin. 2018,13:13-19.

  1. Line 142: add hydrogels as a topical form of melatonin and added an appropriate reference for this claim.

There are several ways to administer melatonin, including tablets, oral solutions, sprays for the nose or oral mucosa, and in the form of skin patches, topical creams or hydrogels [20,21]. 

  1. Soriano, J.L.; Calpena, A.C.; Rodríguez-Lagunas, M.J.; Domènech, Ò.; Bozal-de Febrer, N.; Garduño-Ramírez, M.L.; Clares, B. Endogenous Antioxidant Cocktail Loaded Hydrogel for Topical Wound Healing of Burns. Pharmaceutics. 2020,13:8.

  1. Additionally, to improve our manuscript, in the paragraph Melatonin in skin diseases, we add a table (Table 1.). where we summarized the most common skin disorders in which melatonin can be used. Also, we added a paragraph about supplementary effects of melatonin in allergic diseases in the paragraph Melatonin in allergic skin diseases. Consequently, an old table 1. became table 2.

Table 1. Possible use of melatonin in skin disorders 

Allergic disorders                                      Non-allergic disorders

Atopic dermatitis                                                Melasma

Chronic spontaneous urticaria                          Androgenic alopecia

                                                                          Telogen effluvium

                                                                          Psoriasis

                                                                          Vitiligo

                                                                          Rosacea

                                                                          Acute radiodermatitis

Besides direct effects on sleep cycle, there is possibility that melatonin has also immunomodulatory and anti-inflammatory effects in allergic diseases [46,47]. The activation of the immune system leads to increased production of free radicals associated with decreased melatonin levels and depressed antioxidant enzyme activities in inflammatory skin diseases [47]. As well, in allergic disorders, such as atopic dermatitis and urticaria, mast cells are activated which leads to degranulation of proinflammatory cytokines [47]. Data from experimental studies showed that melatonin reduces inflammation in AD and reduces serum total IgE and IL-4 [47]. But contradictory, in bronchial asthma, allergic disorder of respiratory tract, has pro-inflammatory effect, so its use is not recommended in asthma [47]. Clinical data from experimental studies are still limited.

Reviewer 3 Report

The manuscript cannot accept in journal. Some factors entering into this decision  include depth of research work, novelty, lack of good tables and figures for sections.

Author Response

Dear Reviewer, thank You for Your opinion.

To improve our manuscript, we added a paragraph about additional effects of melatonin in allergic diseases, such as anti-inflammatory and antioxidant effects. Also, we add a table where we summarized the most common skin disorders in which melatonin can be use. I hope that we have at least slightly improved the quality of our manuscript.

Additionally, to improve our manuscript, in the paragraph Melatonin in skin diseases, we add a table (Table 1).  where we summarized the most common skin disorders in which melatonin can be used. Also, we added a paragraph about supplementary effects of melatonin in allergic diseases in the paragraph Melatonin in allergic skin diseases. Consequently, an old table 1. became table 2.

Table 1. Possible use of melatonin in skin disorders 

Allergic disorders                                      Non-allergic disorders

Atopic dermatitis                                                Melasma

Chronic spontaneous urticaria                          Androgenic alopecia

                                                                          Telogen effluvium

                                                                          Psoriasis

                                                                          Vitiligo

                                                                          Rosacea

                                                                          Acute radiodermatitis

Besides direct effects on sleep cycle, there is possibility that melatonin has also immunomodulatory and anti-inflammatory effects in allergic diseases [46,47]. The activation of the immune system leads to increased production of free radicals associated with decreased melatonin levels and depressed antioxidant enzyme activities in inflammatory skin diseases [47]. As well, in allergic disorders, such as atopic dermatitis and urticaria, mast cells are activated which leads to degranulation of proinflammatory cytokines [47].   Data from experimental studies showed that melatonin reduces inflammation in AD and reduces serum total IgE and IL-4 [47]. But contradictory, in bronchial asthma, allergic disorder of respiratory tract, has pro-inflammatory effect, so its use is not recommended in asthma [47]. Clinical data from experimental studies are still limited.

We are at your disposal for further inquiries and changes to improve quality of our manuscript.

Kind regards, authors

Round 2

Reviewer 2 Report

The authors answered all questions comprehensively.

Reviewer 3 Report

I still think that the article is poor in terms of content and novelty, as well as the figures and tables that attract the reader to read, and I still recommend not to publish the article.